# Advantages of Telescopic Screw in Slipped Capital Femoral Epiphysis Treatment: A Retrospective Study and Review of the Literature

**DOI:** 10.3390/children9040469

**Published:** 2022-03-28

**Authors:** Alexandru Ulici, Adelina Ionescu, Diana Dogaru, Olivia Hotoboc, Catalin Nahoi, Cristian Zamfir, Madalina Carp

**Affiliations:** 1Pediatric Orthopedic Surgeon, Department of Pediatric Orthopedic Surgery, Emergency Hospital for Children “Grigore Alexandrescu”, 011743 Bucharest, Romania; alexandruulici@yahoo.com (A.U.); adelina_ionescu@yahoo.com (A.I.); dianna.dogaru@gmail.com (D.D.); hotoboc.olivia@yahoo.com (O.H.); nahoi.catalin@gmail.com (C.N.); dr.zamfircristian@gmail.com (C.Z.); 211th Department, “Carol Davila” University of Medicine and Pharmacy, 050474 Bucharest, Romania

**Keywords:** slipped capital femoral epiphysis, SCFE, Free-Gliding Screw System, children

## Abstract

Background: Slipped capital femoral epiphysis is due to proximal femur physis failure in adolescent patients. Early iatrogenic closure of proximal growth cartilage in children with significant residual growth potential causes complications such as coxa breva, coxa vara, and lower limb length inequalities. The Free-Gliding SCFE Screw System is a self-extending cannulated screw used in Slipped Capital Femoral Epiphysis (SCFE) fixation and femoral neck fractures. Materials and Method: We conducted a retrospective study on 16 patients. All patients under 11 years old were treated by telescopic cannulated screws fixation. The youngest patient was 7 years old. Results: Out of the 22 operated hips, 2 screws have failed, thus resulting in a lack of telescoping of the screw. We discovered an average lengthening of approximately 10 mm at 24 months postoperative check-up in 20 hips in which lengthening took place. According to the Notzli method, none of the patients had an alpha angle value greater than 48 degrees. Conclusion: Fixation with telescopic screw for SCFE in patients less than 11 years old, with mild to moderate slippage, allows the continuous growth and remodeling of the proximal femur, thus avoiding deformities such as coxa breva, coxa vara, FAI, AVN, limb length discrepancies and also allows good range of motion.

## 1. Introduction

Slipped capital femoral epiphysis is due to proximal femur physis failure in adolescent patients generating the displacement of the femoral head on the metaphysis [1]. More recent research shows that the proximal femoral metaphysis slides anterior to the epiphysis, at growth cartilage level, acutely or progressively, causing a varus deformity [2]. Because the symptoms are not specific, such as knee pain and limping, there is approximately an 8-week delay period in diagnosis [1].

An orthopedic physician should have a high suspicion for this disease. The cause for the slip is unknown but has a high incidence in children that are obese or have renal disease, hypothyroidism, and growth hormone deficiency. Advance retroversion of the femoral head may be involved in SCFE [2].

According to the Loder classification, SCFE may be included into two categories: unstable and stable [3]. Unstable forms are those in which the patient had a total loss of function of the affected hip, without being able to walk. Ambulatory patients, with or without walking support, were included in the group of stable forms.

Depending on the time of symptoms onset, SCFE is considered acute if the symptoms start suddenly and last less than 3 weeks anterior to referral, chronic if symptoms persist for at least 3 weeks, or chronic-acute if the patient has a history of painful gait with insidious onset, but with a sudden worsening of symptoms [4,5].

Treatment involves stabilizing the epiphysis and preventing further slipping, as well as complications such as chondrolysis, femoral impingement, and avascular necrosis of the femoral head [2]. The treatment of SCFE can be divided depending on the possibility of walking on the affected limb [3]. For stable SCFE, closed reduction is not recommended. In situ fixation, osteotomy, and femoral-acetabular impingement resection are treatment options [6].

For chronic severe stable cases of SCFE (over 50 degrees Southwick angle), the modified-Dunn treatment is recommended [7]. When this approach is used, care is advised because recent studies have documented a high risk of sequelae, including postoperative femoral head avascular necrosis (AVN) and hip instability. Treatment for unstable SCFE is challenging, and the rate of complications is high. The majority of unstable SCFE cases had previously been treated with a closed technique, making postoperative AVN difficult to anticipate. However, in recent years, the therapy of unstable SCFE has evolved, and open reduction is recommended monitoring the physeal hemodynamics [6,8].

Early iatrogenic closure of proximal growth cartilage in children with significant residual growth potential causes complications such as coxa breva, coxa vara, and lower limb length discrepancies [9]. The epiphysis is frequently fixed with a single screw, and this could lead to premature closure of the growth cartilage. Therefore, the telescopic screws are a better choice for young patients [6]. 

Other therapeutic approaches were documented by Ogden and Southwick, who used an extra-articular bone graft to stimulate closure of the proximal femoral growth cartilage [10]. Segal was the one who advised using a pin or K-wire to prevent injury to the femoral physis [11].

There is a tendency of surgeons to use a dynamic system that allows the remodeling and continuous growth of the proximal femur instead of the rigid system used in the past. Johansson, Nyström, and Knowles pines are mentioned only as of historical importance. The devices currently used for in situ fixing are Kirschner wires, Hansson pins, partially or fully threaded cannulated screws and telescopic screws [12,13].

The telescopic cannulated screw system was developed by a medical equipment manufacturer. It achieves the stabilization of the femoral cervico-cephalic complex by threading both segments. The Free-Gliding SCFE Screw System is a self-extending cannulated screw used in Slipped Capital Femoral Epiphysis (SCFE) fixation and femoral neck fractures. The design of the screw includes a male component (which is attached to the lateral cortex), a female component (which is attached at the proximal epiphysis), and a Cap Figure 1. Anchorage of the components is achieved through threaded ends. After minimal dissection, a guide wire is inserted in the center of the proximal femoral epiphysis under radiological control on both orthogonal views, without exiting the femoral head (at least 3 mm remain to the subchondral bone). A cannulated drill is used to prepare the screw path, without passing beyond the growth cartilage. Using the female driver, the female component is inserted into the epiphysis with all its threads. The male component and the cap are then placed. The stable fixation and rotational stability are created at the fracture (slip) site preventing compression forces thus avoiding premature growth arrest. The end cap does not allow bone overgrowth nor disturbance of the overlying soft tissues and facilitates the extraction of the screw. To ensure that the screw will grow with the femoral neck, the manufacturer’s instructions recommend the complete passage of the threaded female segment in the growth cartilage on both radiological images (anteroposterior and lateral) [14].

## 2. Methods

We conducted a retrospective study on 16 patients who were 11 years old and under that were admitted to our department between January 2015 and November 2020. They had been diagnosed with mild and moderate slipped capital femoral epiphysis and were treated by telescopic cannulated screws fixation. The youngest patient was 7 years old. 

The initial evaluation of the patients involved obtaining demographic data, trauma history, medical records, as well as establishing the moment of symptom onset. No endocrine pathology or other systemic disease was detected in any patient. The clinical examination involved inspection, palpation, assessment of hip joint mobility, and their ability to walk. The laboratory investigations used were blood tests, including inflammatory markers.

All patients were initially evaluated using radiographs of the hip in orthogonal projections (anteroposterior and frog leg view). At follow-ups, the patients were re-evaluated clinically and radiologically.

The epiphyseal slip relative to the proximal femoral metaphysis was established using the Klein line on the anteroposterior X-ray in all patients [15]. Seven patients needed CT scans to confirm the diagnosis. Southwick method was used to establish the degree of slippage [16]. Patients were included in three groups: slight slip below 30 degrees, moderate between 30 and 50 degrees, and severe over 50 degrees [16].

In all patients, fixation was performed using a steel telescopic cannulated screw with a diameter of 6.5 mm with variable lengths that were established intraoperatively. Gentle orthopedic reduction of SCFE was performed, under general anesthesia, prior to fixation in acute cases with the patient placed on the orthopedic table. After progressive traction of the limb, abduction and internal rotation of the affected hip was done. In chronic cases, the “in situ” fixation with a screw was performed without attempting orthopedic reduction.

After surgery, patients did not bear weight on the operated lower limb for 10 days, then resumed progressive gait with support for another 4 weeks. A radiologic evaluation was performed at 24 h, 14 days, 6 weeks, 12 weeks post-op, and then at 6-month intervals until 6 years and a half of follow-up. The minimum follow-up was 20 months with a mean of 48 months. 

The risk of femoral-acetabular impingement was predicted by measuring the alpha angle according to Notzli’s method [17,18].

At the last follow-up, every patient’s skeletal age was evaluated based on their wrist-hand radiograph.

## 3. Results

The study included 16 patients aged 7 to 11 years old, with a mean age of 9 years and 6 months at the time of surgery. BMI was between 24 and 29 with an average of 26.22. The group included 6 female patients and 10 male patients. Out of the 22 hips, 4 were chronic and 18 hips acute.

The results of the study are summarized in Table 1 and Table 2.

We treated a total of 22 hips, 6 children were operated on bilaterally and 10 unilaterally (6 on the right side and 4 on the left side). The Southwick angle was between 15 and 38, with an average of 20 degrees. The degree of displacement was calculated to be moderate for two patients and mild for 14. An average reduction of 5 degrees was obtained on the acute cases.

Free gliding screws with 6.5 mm diameter and length ranging from 50 to 100 mm, with an average length of 76 mm, were used in all patients. An average of four threads was passed to the epiphysis above the growth cartilage. The minimum number of threads was three. 

Out of the 22 operated hips, 2 screws have failed, thus resulting in a lack of telescoping of the screw. We had an average lengthening of approximately 10 mm at 24 months follow-up in 20 hips in which lengthening took place. At the last follow-up, the maximum lengthening was 22 mm, with a mean of 14.6 mm.

At 24 months post-op, the average difference in femoral neck length of the operated hip and the non-operated hips was approximately 3.06 mm. When calculating this difference in the length of the femoral neck, we did not consider the cases of SCFE operated bilaterally. The alpha angle measured according to the Notzli method averaged about 42 degrees. None of the patients had an alpha angle value greater than 48 degrees.

In two cases, at 16 months postoperatively we observed screw failure (entire screw slipped) Figure 2, most likely due to accidental crossings with the drill through the growth cartilage, and of an insufficient number of threads passed through the physis (three). No patient having at least four threads passed over the physis presented with slippage or no telescoping of the screw.

In one case, no lengthening of the screw was noticed 12 months after surgery, but with further follow-ups, the screw extended 15 mm after 3 years Figure 3. The follow-up should be extended to more years depending on the age and growth potential of the patient.

Out of the three patients in which only three threads were passed over the growth cartilage, one had a lengthening of 4 mm; the other two did not telescope.

All cases in our clinic were operated by five doctors with different training and experience.

At the last follow-up all patients were reevaluated clinically and radiographically. We noted their new BMI, Southwick angle, ROM (range of motion), telescoping length, and length difference of the femoral neck. Some of the patients were last seen 3 years prior. No signs of avascular necrosis (AVN) were observed. Furthermore, five of the patients achieved skeletal maturity with no further slippage, impingement, limb length discrepancies, or AVN.

## 4. Discussions

The globally accepted therapeutic surgical choice is proximal epiphysiodesis with the help of a threaded screw inserted percutaneously into the proximal femoral epiphysis [19].

According to Kumm et al., dynamic in situ fixation with a telescopic screw ensures the long-term stability of the head on the femoral neck, thus avoiding the premature closure of the proximal physis and allowing the normal growth of the hip [4]. Slipped capital femoral epiphysis requires surgical management. The surgical technique is chosen according to the severity and type of slippage. Hackenbroch et al. stated that the telescopic cannulated screw is used in cases of mild slippage of the epiphysis (<30 degrees) [20], on youngsters aged 8 to 13, with the best outcomes in individuals as young as 10 years of age [6]. He also used this technique in case of prophylactic fixation of the contralateral femoral head, as well as in case of a moderate slip of the epiphysis, after the closed reduction only if the residual slip does not exceed 30 degrees [20].

The main negative consequence of epiphysiodesis is proximal femoral growth arrest secondary to physis damage by drilling and threaded screw insertion [14].

The shortening of the femoral neck reduces the strength of the abductor muscles of the thigh, thus generating functional impairment of the lower limb in abduction and flexion. Furthermore, the change in hip mechanics generates a phenomenon of femoral-acetabular impingement that can cause early osteoarthritis of the hip in adulthood [21].

Örtegren et al. analyzes the growth potential in 54 pediatric patients diagnosed with SCFE and finds that this suffering of the proximal femur directly causes growth arrest resulting in a length difference of 3 mm between the affected and the healthy side (7 mm vs. 10 mm) [22]. Therefore, we can expect slow progressive growth of the affected femur compared to the contralateral side in case of the telescopic screw usage. 

The manufacturer recommends using a drill with an appropriate size to the screw diameter to prepare the path for the mother component and advance it strictly up to the growth cartilage, not beyond it. This step allows the screw to be attached to the epiphysis through the mother component and telescope, as well as to limit the additional suffering of the growth cartilage [14]. It also mentions that all the threads of the female component must pass the physis. We observed that a minimum of four threads ensure telescoping of the screw.

Femoral-acetabular impingement is a frequent consequence of many SCFE regardless of the degree of slippage [23].

Morash et al. discovered a greater remodeling potential of femoral neck cam deformity in cases of SCFE treated with free gliding screws compared to simple cannulated screws [24]. They had 32 hips in 16 patients treated with free gliding screws, and 102 hips in 55 patients treated with standard screws. The alpha angle remodeled 12.9 degrees for the free gliding screw compared to 4.3 degrees for the standard screw [24]. They also stated that telescopic screws lengthened more in prophylactic operated hips than in SCFE hips. Before this paper, another study published by Örtegren et al. discovered that the use of a system that allows the continuous growth of the femoral neck can reduce the risk of femoral-acetabular impingement [25]. 

It can be considered that the risk of SCFE complication decreases after bone maturation. Skeletal age can be evaluated based on a wrist-hand antero-posterior radiograph [26]. According to Cole et al., the mean age of skeletal maturity in boys is 16.5 years and 15 years in girls [27]. Five out of sixteen patients presented bone maturity at the last radiologic follow-up and no radiologic changes suggesting impingement or AVN. Krahn et al. discovered 36 cases of AVN out of 264 patients developed after SCFE within 18 months from the slippage [28]. In our series, after a minimum of 20 months of follow-up we did not identify any cases of AVN.

The advantage of the telescopic screw used in skeletally immature patients, especially in prophylactically treated hips, is the avoidance of trochanteric overgrowth, coxa breva, and femoral head asphericity to prevent iatrogenic deformity determined by threaded cannulated screws [17,29]. Hansen also reported no case of coxa breva in prophylactic hips treated with dynamic systems such as free gliding screws compared with traditional cannulated screws [30].

Kumm et al. studied 29 hips with slipped capital femoral epiphysis with mild slip (under 30 degrees) treated by dynamic screw fixation. He did not report any case of proximal femoral growth disturbance in his study [20]. 

The average length difference between the operated hip and the contralateral one was 3.06 mm, a functional insignificant value in terms of a possible length discrepancy of the lower limbs caused by iatrogenic epiphysiodesis or lack of screw telescoping. Thus, the use of the telescopic screw allowed the physiological growth of the operated hip compared to the contralateral hip.

Several authors admit in their studies that a value of over 50 degrees of the alpha angle predicts the appearance of cam-type femoral-acetabular impingement [17,31,32]. In our experience, we found a good remodeling of the femoral neck, for all operated patients with free gliding screws with a measured alpha angle below 50 degrees.

Another retrospective study was performed in our clinic, and it included 70 patients and 81 hips treated between 2010 and 2015 for SCFE by in situ fixation with a cannulated screw. It was observed that even in mild and moderate cases there is a high incidence of FAI [33].

We had two cases of moderate slips with Southwick angles of 32 and 38 degrees; the outcome was good even though there was a screw failing in one of the cases. Telescopic screws may be used for in situ fixation for chronic moderate SCFE or after reduction for acute and moderate SCFE. There is need for further study with more moderate cases of SCFE to better understand the evolution of these cases in time.

There are a couple of limitations in the current paper. First, there was a modest number of patients included in the study. Second, the follow-up period was short for some of the subjects compared with others.

We think that this type of treatment should be considered for all SCFE patients under 11 years of age. Further studies and longer follow-up periods are needed to better understand the changes in the proximal femur anatomy in patients treated with telescopic screws. We consider that this study is important because there are not many papers that evaluate patients treated with free gliding SCFE screws.

## 5. Conclusions

Fixation with telescopic screw for SCFE in patients less than 11 years old, with mild to moderate slippage, allows the continuous growth and remodeling of the proximal femur, thus avoiding deformities such as coxa breva, coxa vara, FAI, AVN, and limb length discrepancies and also allows good range of motion.

The telescopic screw technique is relatively simple but has a longer learning curve than the threaded cannulated screws. We encountered two cases of screw failure most probably due to accidentally passing with the drill beyond the proximal femoral physis which led to slippage of the screw. Thus, usage of the telescopic screw needs a good surgical technique. A minimum of four threads of the female component should be passed over the femoral physis. Further studies are needed on this subject.

Because of the scarce number of articles on this subject, the current paper tries to bring forward our experience with this type of fixation. It would be useful to conduct new studies in the future, most importantly, comparative papers about SCFE treatment with free gliding screws versus partially threaded cannulated screws to improve the outcome of young patients with significant growth potential left.

## Figures and Tables

**Figure 1 children-09-00469-f001:**
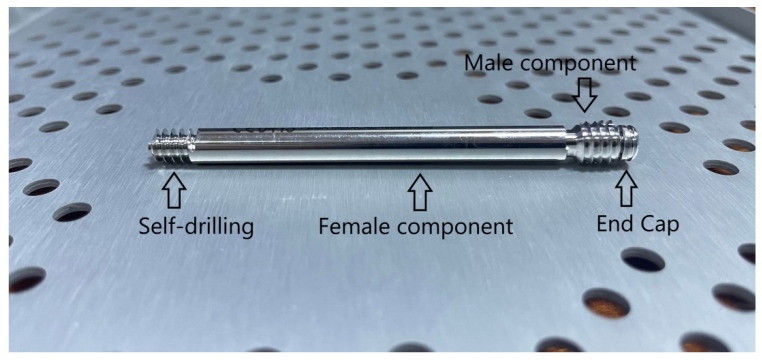
The Free-Gliding SCFE Screw System.

**Figure 2 children-09-00469-f002:**
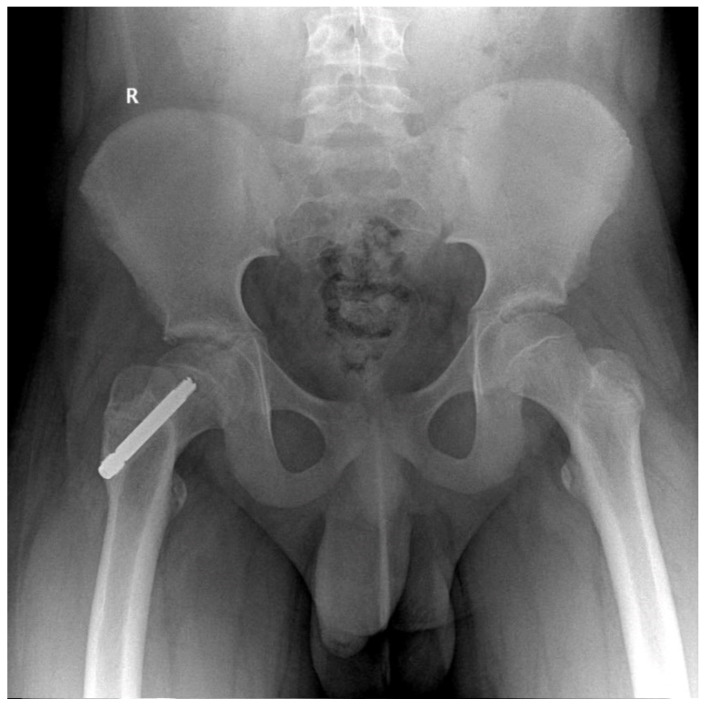
Male patient, 9 years old, slipped screw at 16 months follow-up.

**Figure 3 children-09-00469-f003:**
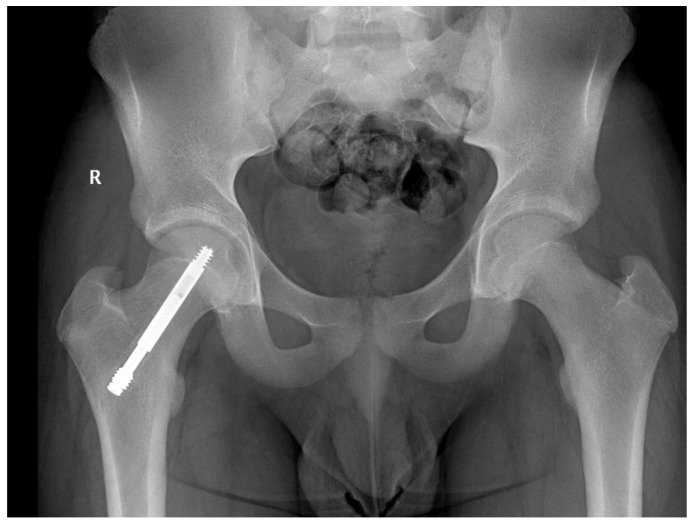
Male patient, 11 years old at time of surgery, screw telescoped at 36 months follow-up; no length discrepancies noted.

**Table 1 children-09-00469-t001:** Summary of Demographic Data and Results; M-male; F-female; L-left; R-right; Ac-acute; Ch-chronic; F-flexion of the hip; IR-internal rotation; ER-external rotation; ADD-adduction; ABD-abduction.

Case No	Gender	Age (Years)	Side	Type	BMI Pre-Op	BMI Follow-Up	Displacement Type	Skeletally Mature	Follow-Up (Months)	ROM Follow-Up
1	F	11	L	Ch	28	26	moderate	No	20	F:100; IR:30; ER:40; ADD:40; ABD:35
2	M	11	R	Ac	29	27	mild	No	23	F:105; IR:40; ER:50; ADD:50; ABD:40
3	F	9	L	Ac	27	27	mild	Yes	76	F:110; IR:40; ER:45; ADD:50; ABD:45
		10	R	Ac	27	27	mild	Yes	64	F: 105; IR:40; ER:45; ADD:45; ABD:45
4	F	8	R	Ac	26	23	mild	No	56	F:110; IR:45; ER:50; ADD:55; ABD:45
		9	L	Ac	27	23	mild	No	49	F:110; IR:45; ER:50; ADD:55; ABD:45
5	M	10	L	Ac	28	26	mild	No	40	F:105; IR:40; ER:45; ADD:45; ABD:40
		10	R	Ac	28	26	mild	No	40	F:105; IR:40; ER:45; ADD:45; ABD:40
6	F	10	R	Ch	27	25	mild	Yes	79	F:100; IR:40; ER:40; ADD:45; ABD:40
7	M	11	R	Ac	26	27	mild	Yes	73	F:105; IR:40; ER:45; ADD:45; ABD:40
		11	L	Ac	26	27	mild	Yes	73	F:110; IR:40; ER:45; ADD:50; ABD:45
8	M	9	L	Ch	26	26	moderate	No	31	F:95; IR:35; ER:45; ADD:40; ABD:35
9	M	10	L	Ac	27	24	mild	No	30	F:105; IR:40; ER:45; ADD:45; ABD:40
		10	R	Ac	27	24	mild	No	30	F:105; IR:40; ER:45; ADD:40; ABD:40
10	M	7	R	Ac	25	24	mild	No	57	F:110; IR:45; ER:50; ADD:55; ABD:45
11	M	10	R	Ch	26	26	mild	No	24	F:100; IR:30; ER:40; ADD:40; ABD:35
12	M	11	R	Ac	25	27	mild	Yes	60	F:115; IR:45; ER:50; ADD:55; ABD:45
13	F	7	R	Ac	24	23	mild	No	43	F:115; IR:45; ER:45; ADD:45; ABD:40
		7	L	Ac	24	23	mild	No	43	F:110; IR:40; ER:45; ADD:45; ABD:40
14	M	9	L	Ac	25	26	mild	No	51	F:110; IR:45; ER:50; ADD:55; ABD:45
15	F	9	L	Ac	24	25	mild	Yes	81	F:110; IR:40; ER:45; ADD:50; ABD:45
16	M	10	R	Ac	25	24	mild	No	25	F:105; IR:40; ER:50; ADD:50; ABD:40

**Table 2 children-09-00469-t002:** Result and Follow-up Data.

Case No	Southwick Angle Pre-Op	Southwick Angle Post-Op	Southwick Angle Follow-Up	Screw Length (mm)	Thread (No)	Lenghtening (mm) Intermediary	Lenghtening (mm) Follow-Up	Diff Fem Neck (mm) Intermediary	Diff Fem Neck (mm) Follow-Up	A V N	Telescoped	Failure of the Screw
1	38	38	38	80	3	0	0	10	11	No	No	Yes
2	20	14	14	100	5	9	10	1	1	No	Yes	No
3	17	12	10	80	5	12	17	0	0	No	Yes	No
	19	13	10	84	5	11	15	0	0	No	Yes	No
4	28	22	20	98	4	14	17	1	0	No	Yes	No
	26	18	16	100	4	12	14	0	0	No	Yes	No
5	15	10	10	78	4	12	16	3	0	No	Yes	No
	15	10	10	76	5	13	16	0	0	No	Yes	No
6	18	18	14	72	5	12	21	1	1	No	Yes	No
7	15	10	8	80	5	9	20	3	0	No	Yes	No
	20	14	10	82	4	12	18	0	0	No	Yes	No
8	32	32	32	68	4	15	16	0	0	No	Yes	No
9	26	20	20	50	5	12	14	2	2	No	Yes	No
	20	16	16	72	5	14	16	0	0	No	Yes	No
10	16	10	10	74	3	15	20	9	5	No	Yes	No
11	17	17	17	80	4	12	14	0	0	No	Yes	No
12	18	14	16	70	5	0	0	10	15	No	No	Yes
13	16	12	10	68	3	4	8	7	0	No	Yes	No
	15	12	10	70	4	14	16	0	0	No	Yes	No
14	20	14	12	72	5	12	18	2	0	No	Yes	No
15	16	10	10	68	4	14	22	0	0	No	Yes	No
16	18	14	14	70	5	14	14	0	0	No	Yes	No

## Data Availability

Data available on request.

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
