# Peer review of "Advantages of Telescopic Screw in Slipped Capital Femoral Epiphysis Treatment: A Retrospective Study and Review of the Literature"

_children, 2022, doi:10.3390/children9040469_

Round 1
Reviewer 1 Report
Actually a very interesting paper with an important message. I would only advise to add an information about the SCFE itself. For the surgeon it is a very interesting message, that also moderate slip is resulting in a good manner. As I can understand the degree of the slip is also not important for the failure rate- therefore I would encourage the authors to underline, that this technique is also fine in moderate slip. Additionally, I would also encourage the authors to emphasize, that failure rate is based on mechanical problems (which is understandable in a procedure, which needs good practice).
Author Response
Thank you for taking the time to review our paper and for your nice words.
We have added more information about SCFE and ethology of SCFE in the introduction section to better describe this disease.
Thank you for pointing out the data about the moderate slips. We have commented about this in the discussion section. Our results were good in moderate slips but, unfortunately, we only had 2 such cases thus we think that more data is needed to better understand the evolution of these patients treated with telescopic screws.
Concerning the failure rate, this is indeed related to mistakes in the procedure and we have emphasised this in the conclusion section for better understanding. It is important for all the recommended steps to be followed in this surgical technique for optimal results.
Reviewer 2 Report
48-55:for severe stable?...or instable???...Here it's not so clear, you have to explain better this items
79-82 you have to explain better the surgical tecnique87-93 you have to summarize thhese items, there are repetitions95...gait analysis of limping children with acute SCFE?
99...lateral: do you mean axial?
103 Why CT scan instead of MRI?
for which reason a radiological evoluation at 14 days?106 what kind of telescopic screw did you used?
I think you can improve the graphic of the tables
Author Response
Thank you for taking the time to revise our manuscript and for the insightful comments.
Comment 1 -48-55:for severe stable?...or instable???...Here it's not so clear, you have to explain better this items
We have modified in the text, we referred to severe stable cases of SCFE
Comment 2- 79-82 you have to explain better the surgical tecnique87-93 you have to summarize these items, there are repetitions95...gait analysis of limping children with acute SCFE?
Thank you for these suggestions. As pointed out we have rewritten part of the text to better explain the surgical technique at lines 80-86.
We have removed a paragraph and summarized data as indicated in the beginning of the material and methods section.
Regarding the gait analysis this was a poor choice of expression, we wanted to say that we evaluated the ability to walk. We have corrected that in the text.
Comment 3- 99...lateral: do you mean axial?
Thank you for pointing this out it was a translated related mistake and we have corrected to frog leg view.
Comment 4- 103 Why CT scan instead of MRI?
Seven of our patients needed a CT scan for diagnosis, many of these cases presented in our clinic with a CT scan recommended in other medical units, probably due to lack of radiological signs and persisting symptoms as well as easier access to CT scans.
Comment 5 for which reason a radiological evaluation at 14 days?106 what kind of telescopic screw did you used?
This was our protocol for follow up. At 14 days post-op the patients return to our clinic for suture removal as well as clinical and radiological evaluation. Considering that this a new method of treatment, we wanted to check the stability of the screws.
Comment 6 I think you can improve the graphic of the tables
We tried to improve the aspect of the tables.
This manuscript is a resubmission of an earlier submission. The following is a list of the peer review reports and author responses from that submission.